

# Individuality and stability of the koala (*Phascolarctos cinereus*) faecal microbiota through time

Raphael Eisenhofer[1,2,*], Kylie L. Brice[3,*], Michaela DJ Blyton[4], Scott E. Bevins[3,†], Kellie Leigh[5], Brajesh K. Singh[3,6], Kristofer M. Helgen[7,8,9], Ian Hough[9], Christopher B. Daniels[9], Natasha Speight[10] and Ben D. Moore[3]

[1] School of Biological Sciences, The University of Adelaide, Adelaide, South Australia, Australia
[2] Australian Research Council Centre of Excellence for Australian Biodiversity and Heritage, The University of Adelaide, Adelaide, South Australia, Australia
[3] Hawkesbury Institute for the Environment, Western Sydney University, Richmond, New South Wales, Australia
[4] School of Chemistry and Molecular Biosciences, Faculty of Science, University of Queensland, Brisbane, Queensland, Australia
[5] Science for Wildlife Ltd, Sydney, New South Wales, Australia
[6] Global Centre for Land Based Innovation, Western Sydney University, Penrith, New South Wales, Australia
[7] Australian Museum Research Institute, Sydney, New South Wales, Australia
[8] Australian Research Council Centre of Excellence for Australian Biodiversity and Heritage, University of New South Wales, Sydney, New South Wales, Australia
[9] Koala Life Foundation, Cleland Wildlife Park, Department for Environment and Water, 365c Mt Lofty Summit Road, Adelaide, South Australia, Australia
[10] School of Animal and Veterinary Sciences, The University of Adelaide, Adelaide, South Australia, Australia
* These authors contributed equally to this work.
† Deceased

Corresponding authors
Raphael Eisenhofer,
raph.eisenhofer@gmail.com
Ben D. Moore,
B.Moore@westernsydney.edu.au

## ABSTRACT

Gut microbiota studies often rely on a single sample taken per individual, representing a snapshot in time. However, we know that gut microbiota composition in many animals exhibits intra-individual variation over the course of days to months. Such temporal variations can be a confounding factor in studies seeking to compare the gut microbiota of different wild populations, or to assess the impact of medical/veterinary interventions. To date, little is known about the variability of the koala (*Phascolarctos cinereus*) gut microbiota through time. Here, we characterise the gut microbiota from faecal samples collected at eight timepoints over a month for a captive population of South Australian koalas (*n* individuals = 7), and monthly over 7 months for a wild population of New South Wales koalas (*n* individuals = 5). Using 16S rRNA gene sequencing, we found that microbial diversity was stable over the course of days to months. Each koala had a distinct faecal microbiota composition which in the captive koalas was stable across days. The wild koalas showed more variation across months, although each individual still maintained a distinct microbial composition. Per koala, an average of 57 (±16) amplicon sequence variants (ASVs) were detected across all time points; these ASVs accounted for an average of 97% (±1.9%) of the faecal microbial community per koala. The koala faecal microbiota exhibits stability over the course of days to months. Such knowledge will

![PeerJ]

be useful for future studies comparing koala populations and developing microbiota interventions for this regionally endangered marsupial.

## INTRODUCTION

Temporal dynamics of the gut microbiota of wild animals are generally poorly understood, due primarily to the difficulty of repeatedly collecting samples from known individuals over an extended period (*David et al., 2014b*; *Faust et al., 2015*). Recently there has been a surge in time-series investigations in wild animals and humans (*Faust et al., 2015*; *Caporaso et al., 2011a*; *David et al., 2014a*; *Ren et al., 2016*; *Amato et al., 2015*; *Gomez et al., 2016*). Most longitudinal studies in wild animals have focused on primates, with diets encompassing a wide variety of plant species, and incorporating multiple plant parts (*e.g.*, foliage, bark, and fruit) in their diet. These studies have included western lowland gorillas (*Gorilla gorilla gorilla*) and mountain gorillas (*Gorilla beringei beringei*) (*Gomez et al., 2016*), black howler monkeys (*Alouatta pigra*) (*Amato et al., 2013*), and Udzungwa red colobus monkeys (*Procolobus gordonorum*) (*Barelli et al., 2015*). Others have focused on primates with more omnivorous diets, including chimpanzees (*Pan troglodytes schweinfurthii*) (*Degnan et al., 2012*), baboons (*Papio cynocephalus*) (*Ren et al., 2016*), ring-tailed lemurs (*Lemur catta*) (*Fogel, 2015*). The results of these studies, particularly those focusing on wild herbivores, have shown that seasonal shifts in food availability (*e.g.*, of ripe fruit) can drive gut microbiota composition. In contrast, studies in humans have found that diet and host lifestyle influence the gut microbiota at a finer (*e.g.*, daily) scale (*David et al., 2014b*, *2014a*; *Faith et al., 2013*). Overall, temporal variability of the gut microbiota varies from animal to animal and can be heavily influenced by diet.

The koala (*Phascolarctos cinereus*), a specialist folivore, has recently been listed as endangered due to threatening processes including habitat fragmentation (*Melzer et al., 2000*), climate change (*Seabrook et al., 2011*), and disease (*e.g.*, chlamydiosis and koala retrovirus) (*Polkinghorne, Hanger & Timms, 2013*; *Simmons et al., 2012*; *McEwen et al., 2021*). The koala is an obligate dietary specialist (*Shipley, Forbey & Moore, 2009*) which relies almost exclusively on eucalypt foliage (from the genera *Eucalyptus*, *Corymbia* and *Angophora*, family Myrtaceae) (*Hindell, Handasyde & Lee, 1985*). Eucalypt foliage is a chemically complex and nutritionally challenging food source (*Moore & Foley, 2005*) and its consumption and digestion is thought to be facilitated by the fermentative ability of gut microbiota and by the host's physiological capacity to detoxify and eliminate plant toxins. Given recent calls that the gut microbiota could be linked to host health and conservation (*Redford et al., 2012*; *Trevelline et al., 2019*), there has been a surge of interest in understanding the role of the gut microbiota in koala diet choice and health. Recently *Brice et al. (2019)* found that within a single koala population, the gut microbiota of koalas differed according to which of two *Eucalyptus* species, manna gum (*Eucalyptus viminalis*)

and messmate (*E. obliqua*), were eaten. They concluded that the koala gut microbiota adjusts to individual diet, presumably enabling optimal degradation and extraction of available nutrients. This population of koalas had experienced a period of exponential growth, resulting in overbrowsing and die-back of their preferred food species, manna gum (*Whisson et al., 2016*). In 2013, hundreds or thousands of koalas died of starvation, due in part to their reluctance to feed on available messmate patches (*Whisson et al., 2016*). In a captive experiment, *Blyton et al. (2019)* altered the microbial community composition of manna gum-feeding koalas to resemble more closely that of messmate-feeding koalas, using faecal inoculations. Individuals with more substantially altered microbiota subsequently increased their consumption of messmate. This demonstrated that an individual koala's gut microbiota influences feeding behaviour and suggests that the microbiota might limit the ability of koalas to feed from messmate during mass starvation events. However, we currently lack a detailed understanding of the stability of the koala gut microbiota through time, particularly in wild koalas. Such knowledge could be useful for disentangling the effects of diet and other treatment effects from natural variations in koalas.

To fill this gap, we characterised the gut microbiota from faecal pellets collected over a one-month period from wild koalas rescued from extensive wildfires that occurred on Kangaroo Island, South Australia (SA), from December 2019–January 2020, and rehomed at Cleland Wildlife Park, SA, as well as from mainland South Australian koalas housed at Cleland. We also undertook a 7-month time-series study to determine the stability of the faecal microbiota of wild koalas living in a mixed eucalypt forest at Mountain Lagoon, in and adjoining Wollemi National Park, NSW, Australia. We aimed to address the questions: (1) how much intra-individual variation in faecal microbiota diversity and composition is there for koalas over days and months, and (2) is there significant inter-individual variation in gut microbial community composition between koalas?

## EXPERIMENTAL PROCEDURES

### Study sites and faecal collection methods

#### Cleland koala population

Pellets were collected from Cleland captive koalas (*n* = 4) and Kangaroo Island koalas (rescued wild koalas, *n* = 3) housed at the Cleland Wildlife Park (34°58′01.5″S, 138°41′49.0″E) in the Adelaide Hills, SA, Australia (Table 1). The koalas were housed individually for the duration of the study, as they were being kept isolated during testing for koala retrovirus and *Chlamydia*. Between April and May 2020, freshly defecated faecal pellets were collected every 3 to 4 days for a period of one month into 5 mL tubes and immediately stored in a −20 °C freezer (Table S2). All koalas were supplied with branches of a range of eucalypt species daily, primarily including river red gum (*E. camaldulensis*), manna gum (*E. viminalis*), and South Australian blue gum (*E. leucoxylon*) (Table S2). Cleland sampling was conducted with University of Adelaide Animal Ethics Committee Approval S-2016-169 and Department for Environment and Water scientific permit Y26054.

**Table 1 Characteristics of koalas in this study.**

| Koala name | Sex | Geographic origin | Entry date (Cleland) | Age (years) |
|---|---|---|---|---|
| Lola | Female | Adelaide Hills, SA | Unknown | 8 |
| Byron | Male | Kangaroo Island, SA | 13/1/2020 | 4 |
| Blaze | Male | Kangaroo Island, SA | 22/1/2020 | 3 |
| Claudette | Female | Kangaroo Island, SA | 22/1/2020 | 4 |
| Leo | Male | Reared at Cleland, SA | NA | 2 |
| George | Male | Reared at Cleland, SA | NA | 2 |
| Theo | Male | Reared at Cleland, SA | NA | 1 |
| Cin | Female | Mountain Lagoon, NSW | NA | Adult |
| Phasco | Female | Mountain Lagoon, NSW | NA | Adult |
| Reus | Female | Mountain Lagoon, NSW | NA | Adult |
| Tos | Male | Mountain Lagoon, NSW | NA | Adult |
| Uno | Male | Mountain Lagoon, NSW | NA | Adult |

### Mountain Lagoon koala population

Faecal pellets were collected from three female and two to male radio-collared koalas (VHF single stage transmitter, Titley Scientific, Brendale, QLD, Australia) that were part of a larger study at Mountain Lagoon (33°10′02″S, 150°28′11″E) in the Blue Mountains, NSW, Australia (Fig. 1). This was a medium-density koala population, inhabiting continuous mixed eucalypt forest, in and adjoining Wollemi National Park. Mountain Lagoon is a small basin with a layer of relatively more-fertile Wianamatta shale overlying Triassic Hawkesbury sandstone (Robbie & Martin, 2007). A range of vegetation communities occur at the site, including Lower Blue Mountains Exposed Red Bloodwood Forest, Blue Mountains Grey Gum-Stringybark Transition Forest, Blue Mountains Blue Gum-Turpentine Gully Forest, Sydney Hinterland Sheltered Turpentine-Apple Forest, and Blue Mountains Shale Cap Forest.

Koalas were recorded using at least 13 different eucalypt species from these five vegetation communities (Gallahar, Leigh & Phalen, 2021) across the genera *Eucalyptus* (subgenera *Symphyomyrtus* and *Eucalyptus*—previously *Monocalyptus*), *Corymbia* and *Angophora* (See Table S1). The wide variety of potential food species available at Mountain Lagoon increases the diversity of digestive and nutritional challenges posed to koalas, as well as exposure to plant secondary metabolites including polyphenols such as tannins, flavonoids and quinic acid derivatives (Marsh et al., 2017), terpenoids (Boland, Brophy & House, 1991), formylated phloroglucinol compounds (FPCs) (Moore et al., 2004) and unsubstituted B-ring flavanones (UBFs) (Marsh et al., 2019). Each of the koalas sampled in the study were recorded using from five to nine different tree species, and predominantly utilised the shale and shale-sandstone transition soil types (Gallahar, Leigh & Phalen, 2021).

The koalas were radio tracked diurnally over a 3-day period at fortnightly intervals between January and July 2015; upon observation of the koala, the occupied tree was identified to species, and the location recorded with a hand-held GPS unit. A mat was
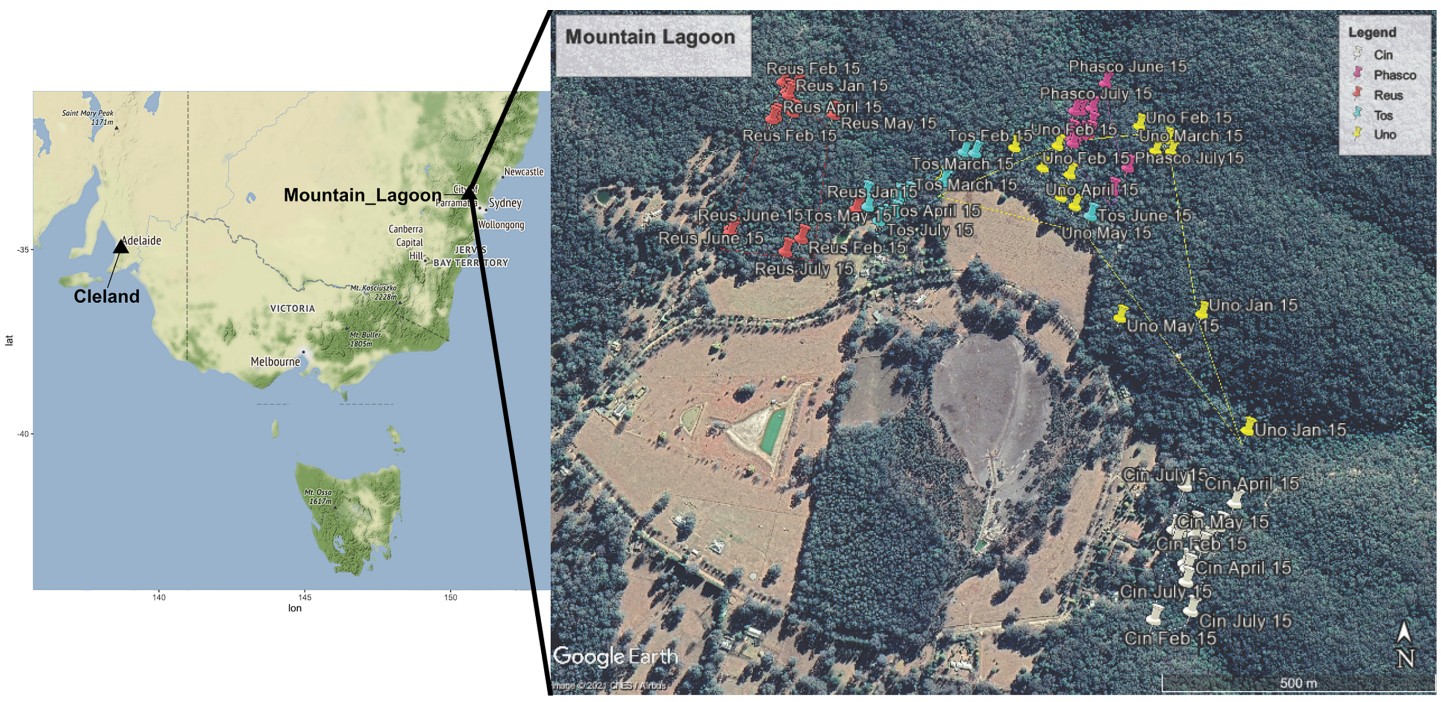

**Figure 1** **Map showing locations of Mountain Lagoon, NSW and Cleland Wildlife Park, SA.** Wild Mountain Lagoon koalas were radio tracked over a 7-month period during 2016. The cutaway map demonstrates the approximate home range (100% minimum convex polygon) for each koala at Mountain Lagoon. Each coloured pin represents one tracking time point for a particular koala. Key: White Cin (female), pink Phasco (female), red Reus (female), blue Tos (male), and yellow represents Uno (male). Map data ©2021 Google CNES/Airbus.

placed directly underneath the koala to catch fresh faecal pellets and periodically checked throughout the day at typical intervals of 15 min (no greater than 4 h). If no pellets were observed during the day, the mats were left out overnight and pellets collected the following morning. Collected pellets were bagged and placed on ice until they could be stored in a −20 °C freezer. Studies into the impact of post-collection storage, handling and DNA extraction methods on DNA quality, microbial composition and diversity of faecal samples have concluded that there is no significant impact of short-term storage in temperatures of 4 °C for up to 24–72 h. They also found short or long-term storage at −20 and −80 °C did not have a significant impact (*Carroll et al., 2012*; *Ezzy et al., 2019*; *Lauber et al., 2010*). The Mountain Lagoon project was conducted as part of a koala project with the approval of the Western Sydney University Animal Care and Ethics Committee (ACEC), reference number: A10670 and New South Wales National Parks and Wildlife Service Scientific Licence SL101364.

## DNA extraction

Microbial DNA was extracted from faecal pellets using the DNeasy PowerSoil® DNA isolation kit (QIAGEN) as per the manufacturer's instructions. Briefly, 0.25 grams (g) of faeces (frozen) was added to the 750 microliters (μl) bead bed, 100 μl of sterilised milli-Q ultra-pure water and 160 μl of the C1 solution were added and beat-beating was performed by vortexing the tubes on high for 10 minutes (min). Final elution was performed with
100 µl of sterile nuclease-free water (Invitrogen, Mulgrave, VIC, Australia). DNA concentration was quantified by Qubit 2.0 Fluorometric assay (Life Technologies, Carlsbad, CA, USA), and adjusted to a concentration of 10 ng/µl.

## 16S rRNA gene library preparation

### Cleland samples

Single PCR reactions (*Marotz et al., 2019*) were performed for all extracted Cleland koala faecal DNA using the reverse-barcoded V4 region 515F-806R primer pair (515F 5-GTGCC AGCMGCCGCGGTAA-3 and 806R 5-GGACTACHVGGGTWTCTAAT-3) (*Caporaso et al., 2011b*). Each reaction contained 2.5 µl HiFi Buffer (X10), 0.1 µl Platinum$^{TM}$ Taq DNA Polymerase (ThermoFisher, Waltham, MA, USA), 1µl MgSO4 (50 mM), 0.2 µl dNTP mix (100 µM), 0.5 µl (10 µM) of each primer, 1 µl template DNA, reaction volumes were adjusted to a total of 25 µl with milli-Q ultra-pure water. PCR conditions consisted of 94 °C (3 min), 94 °C (45 seconds (s)), annealing temperature of 50 °C (60 s), extension temperature of 68 °C (90 s), repeated 35 times, then 68 °C (10 min).

### Mountain Lagoon samples

The V4 region of the bacterial 16S rRNA gene was amplified using primers 515F and 806R from *Caporaso et al. (2011b)*. Each 20 µl reaction contained 10 µl MyTaq$^{TM}$ HS Red Mix (2×) hot-start PCR (BioLine, Taunton, MA, USA), 1 µl of each primer (20 micromole (µM); Applied Biosystems®/Life Technologies Australia Pty, Ltd, Mulgrave, VIC, Australia), and 2 µl template DNA (20 nanogram (ng)/µl). Reaction volumes were adjusted to a total of 20 µl with milli-Q ultra-pure water (Millipak® 40 Millipore; Merck Millipore, Kilsyth, VIC, Australia). PCR conditions were 95 °C (1 min), 95 °C (30 seconds (s)), annealing temperature of 61.5 °C (30 s), extension temperature of 72 °C (1 min), repeated 35 times, then 72 °C (10 min). It should be noted that two different polymerases and annealing temperatures were used in this study for the different groups of koalas (Cleland and Mountain Lagoon). While polymerase choice can introduce variation, it has been demonstrated that the amount of variation between individual human faecal samples is far greater than the variation introduced by polymerase choice (*Sze & Schloss, 2019*). Additionally, the present study was focused on differences within individuals and populations, in which there were no differences in polymerase or annealing temperatures.

Paired-end DNA sequencing was performed on an Illumina MiSeq® platform, using the MiSeq reagent Kit v3 (Mountain Lagoon) and v2 (Cleland) by the Next Generation Sequencing Facility at Western Sydney University, Richmond, NSW (Mountain Lagoon samples) and at the SAHMRI (South Australian Health and Medical Research Institute, Adelaide, South Australia), SA (Cleland samples).

## Bioinformatic analysis

Analysis of all sequence data was performed using Quantitative Insights into Microbial Ecology (QIIME2) pipeline, version 2020.6 (*Bolyen et al., 2019*) unless otherwise stated. Forward read sequences were denoised using Deblur (*Amir et al., 2017*) to create a BIOM table (*McDonald et al., 2012*). Taxonomy was assigned using a naive Bayesian classifier
against the SILVA 138 v4 database (*Bokulich et al., 2018*; *Quast et al., 2013*). A phylogenetic tree was constructed using the SATe-Enabled Phylogenetic Placement (SEPP) (*Mirarab, Nguyen & Warnow, 2012*) insertion pipeline as implemented in the q2-fragment-insertion-plugin (*Janssen et al., 2018*; *Matsen, Kodner & Armbrust, 2010*), with the SILVA 128 reference tree.

Count tables and diversity metrics were imported into R Studio (*RStudio Team, 2015*) and phyloseq (*McMurdie & Holmes, 2013*) and manipulated using dplyr (https://github.com/tidyverse/dplyr), tidyr (https://github.com/tidyverse/tidyr), and the microbiome package (https://github.com/microbiome/microbiome). We did not filter rare ASVs from our dataset. We tested filtering ASVs with a relative abundance <0.25%, but this did not change the original biological interpretations of the results. This supplementary analyses can be found in a markdown document here (https://github.com/EisenRa/2022_Koala_Faeces_R). We set the random subsampling depth to 36,622 to retain all samples and computed various alpha and beta diversity metrics using the microbiome R package, including richness, pielou's evenness (*Pielou, 1966*), unweighted (*Lozupone & Knight, 2005*) and weighted (*Lozupone et al., 2007*) UniFrac distances. Plots were created using ggplot2 (*Wickham, 2016*). Alpha diversity ANOVAs (*Anderson, 2001*) and linear models were run using the lme4, lmerTest, and stargazer R packages. Beta diversity PERMANOVAs (*Anderson, 2017*) were run using the vegan R package. All QIIME2 and R code used for analysis and visualisation are available at the following GitHub repository (https://github.com/EisenRa/2022_Koala_Faeces_R).

## RESULTS

DNA sequencing of the 80 koala faecal samples produced 17,457,412 reads (mean 218,218 reads per sample, min 65,214, max 1,009,351), which after de-noising yielded a total of 947 ASVs.

### Temporal stability of koala faecal microbiota diversity through time

To measure the stability of the koala gut microbiota through time, we longitudinally sampled faecal samples from koalas at the level of days and months. Koalas housed at Cleland, South Australia exhibited stable gut microbial evenness and richness (despite subtle variation in *Eucalyptus* species eaten; Tables S2 and S3) when sampled every 4 days over a month (Figs. 2A and 2C). The one exception was the koala named Lola, who experienced a rise and subsequent stabilisation of microbial richness from days 16 to 24. Wild Mountain Lagoon koalas also generally exhibited stable evenness and richness when sampled monthly for 6 months (Figs. 2B and 2C). There were significant differences among koalas in microbial diversity (ANOVA of richness $p$-value < 0.001, ANOVA of evenness < 0.001; Fig. S1), and Mountain Lagoon koalas had significantly lower ASV richness (but not evenness) compared to Cleland koalas (mean richness Mountain Lagoon = 112 ± 24.3, mean richness Cleland = 145 ± 29.5; $t = -2.716$, $p = 0.0203$; Fig. S2). However, given the overall low population-level sample sizes (seven $vs$ five), this result should be met with caution.

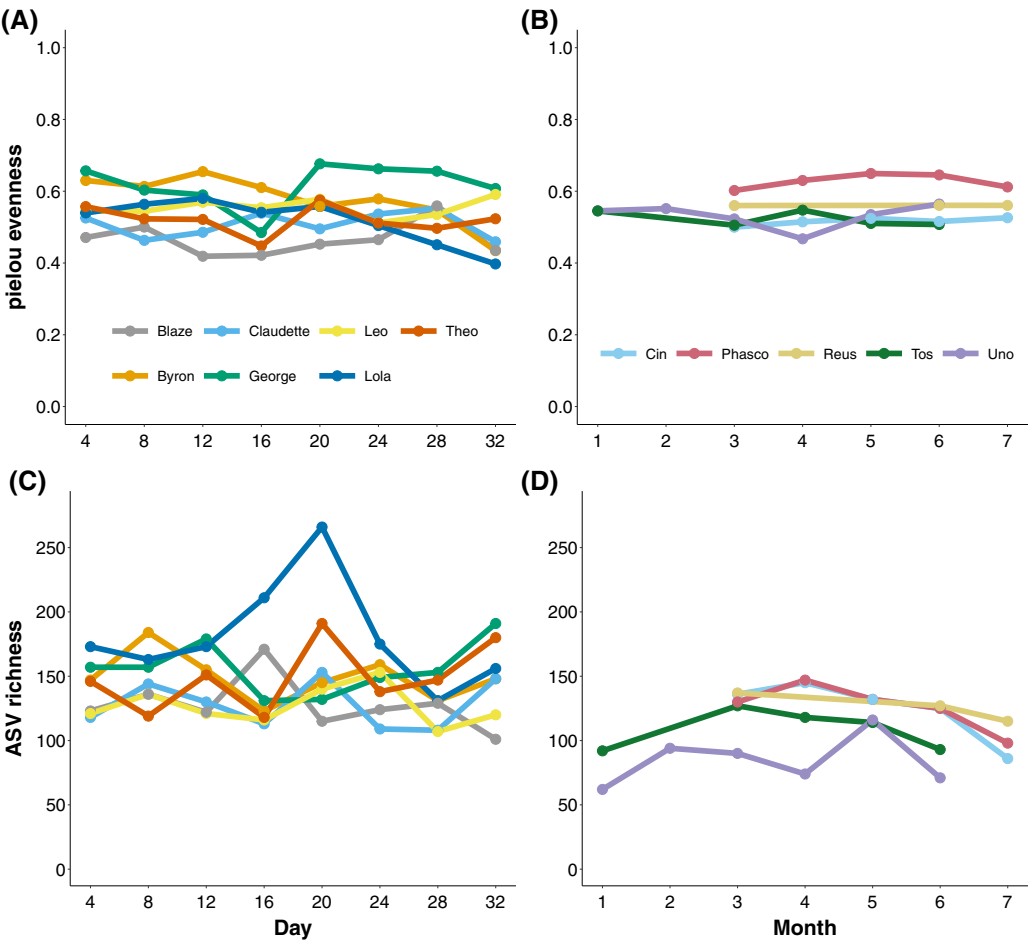

**Figure 2 Microbial evenness and richness in koala faecal samples.** Microbial evenness and richness in koala faecal samples through time from days (Cleland; (A) and (C)) to months (Mountain Lagoon; (B) and (D)).

## Faecal ASV stability through time at the individual level

The dominant members of the koala faecal microbiota remained highly stable at the individual level (Table 2). Each koala had on average 249 (±43) ASV detected over the course of sampling. Of these, an average of 57 (±16) were detected at all time points per koala. These ASVs accounted for on average 97% (±1.9%) of the relative abundance of ASVs in each koala, suggesting a stable individual core microbiota at the ASV level for koalas over the course of days to months. Within the Cleland koalas, five ASVs were found in all koalas at all time points, of which one (classified as family = *Tannerellaceae*, genus = *Parabacteroides*) had a mean relative abundance of 18.6% (±15.3%). Nine ASVs were found in all Mountain Lagoon koala samples, two of which (classified as family = *Paludibacteraceae*, and classified as family = *Tannerellaceae*, genus = *Parabacteroides*) had mean relative abundances of 15.6% (±9.9%) and 20.7% (±11.2%), respectively (Table S4). While no ASVs were shared between all koalas in the study (*i.e.*, Cleland and Mountain Lagoon), 82 of 947 ASVs were shared between Cleland and Mountain Lagoon (with 501 and 353 being found only in Cleland and Mountain Lagoon, respectively; Fig. S2). These

**Table 2  ASV-level statistics per koala.**

| Koala | ASVs per koala | Individual core ASVs | Mean rel. abun. individual core ASVs |
|---|---|---|---|
| Blaze (CL) | 259 | 58 | 0.99 |
| Byron (CL) | 286 | 67 | 0.93 |
| Cin (CL) | 234 | 43 | 0.97 |
| Claudette (CL) | 256 | 45 | 0.98 |
| George (CL) | 279 | 84 | 0.97 |
| Leo (CL) | 233 | 59 | 0.99 |
| Lola (CL) | 340 | 76 | 0.98 |
| Phasco (ML) | 206 | 62 | 0.98 |
| Reus (ML) | 204 | 68 | 0.96 |
| Theo (ML) | 284 | 59 | 0.93 |
| Tos (ML) | 200 | 40 | 0.98 |
| Uno (ML) | 204 | 27 | 0.98 |

Note:
'Total ASVs per koala' represents the total number of ASVs detected for each koala through the course of the study. 'Individual core ASVs' is the number of ASVs found at all time points for a given koala. 'Mean rel. abun. individual core ASVs' represents the mean relative abundance of the core ASVs for each koala. CL, Cleland; ML, Mountain Lagoon.

shared ASVs accounted for a mean relative abundance of 36% and 25% for Cleland and Mountain Lagoon koalas, respectively.

## Stability and individuality of koala faecal microbiota composition through time

The analysis of microbial composition by principal coordinates analysis (PCoA) of unweighted and weighted UniFrac distances, indicated that faecal microbial composition tended to cluster by individual (Fig. 3). Over the course of days, Cleland samples showed tight clustering of microbial community composition by individual koala (Fig. 3A; PERMANOVA $p$-value < 0.001, $R^2$ = 0.615), with looser clustering when taking abundance into account (Fig. 3B; PERMANOVA $p$-value < 0.001, $R^2$ = 0.746). In wild koalas sampled over months, the clustering of microbial communities was less tight but still significant (Figs. 3C and 3D; PERMANOVA of unweighted UniFrac $p$-value < 0.001, $R^2$ = 0.506; PERMANOVA of weighted UniFrac $p$-value < 0.001, $R^2$ = 0.757).

Taxonomically we observed similar trends, with Cleland koalas being clearly distinguishable from each other by the types and relative abundances of microbial families, with minor variation in the relative abundances of microbes on the day-to-day scale (Fig. 4). The phylum level composition of the koalas in this study were similar to those of previous koala faecal microbiota studies, being dominated by Bacteroidota and Firmicutes (Fig. S4) (*Brice et al., 2019*; *Littleford-Colquhoun et al., 2022*). Mountain Lagoon koalas were less distinguishable between each other at the family level, except for the individual Phasco, who had much lower levels of *Paludibacteraceae* (Fig. 4) and was also the only koala to demonstrate a strong preference for occupying *Eucalyptus deanei* (K. Leigh, unpublished data 2015). Across all koalas in the study, *Tannerellaceae* had the highest relative abundance (mean 25.7% in Cleland, 36.3% in Mountain Lagoon). Cleland koalas

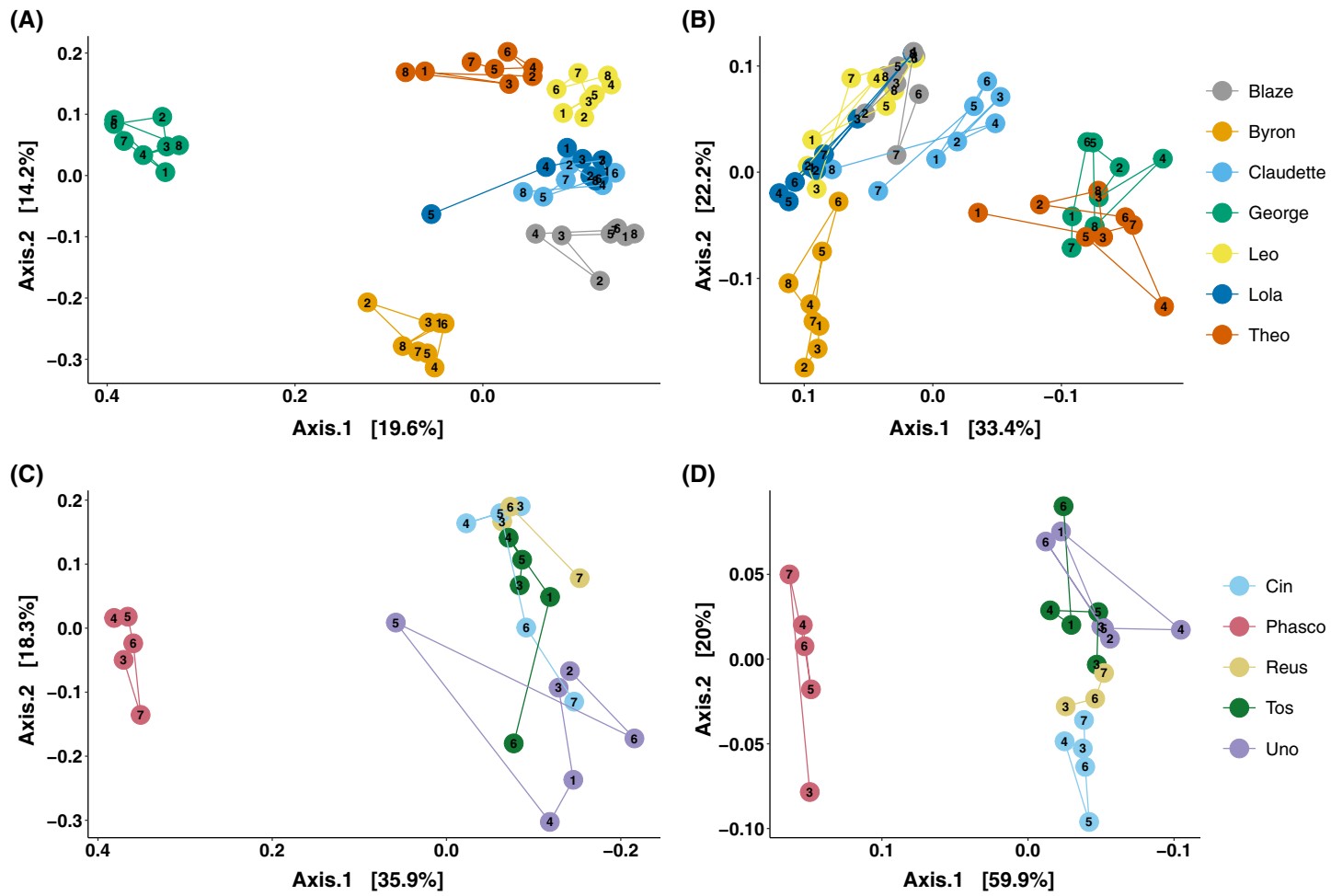

**Figure 3 Principal coordinates analysis of koala faecal microbiota beta diversity.** Principal coordinates analysis (PCoA) of unweighted (A) and weighted (B) UniFrac diversity in the faecal microbiota of Cleland koalas sampled every 4 days for a month, and Mountain Lagoon koalas sampled monthly for 6 months: unweighted (C), weighted (D). Lines connect faecal samples from the same koala in the order they were sampled. Numbers inside points represent the sample collection number.

had higher relative abundances of *Ruminococcaceae* compared to Mountain Lagoon koalas (mean 14.2% *vs* 3.1%, respectively), and *Gastranaerophilales* (mean 6.2% *vs* 0.09%, respectively). *Muribaculaceae* (formerly S24-7) (mean 11.4%), *Akkermansiaceae* (mean 1.5%), and *Prevotellaceae* (mean 1.1%) were detected in Cleland koalas but not Mountain Lagoon koalas. Mountain Lagoon koalas had substantially higher relative abundances of both *Paludibacteraceae* (15.7% *vs* 0.0008%) and *Synergistaceae* (mean 11.5% *vs* 0.9%) compared to Cleland samples. Across all koala samples, the genus *Lonepinella* had a prevalence of 82.5% and a mean relative abundance of 1.1% (min = 0%, max = 9.7%).

## DISCUSSION

The extent of intra-individual variation in faecal microbiota diversity and composition is important to consider, especially when trying to understand factors that drive microbiota changes (*e.g.*, diet or experimental treatment). Given that intra-individual variability is species-specific (*David et al., 2014b*; *Amato et al., 2015*) and potentially context-specific

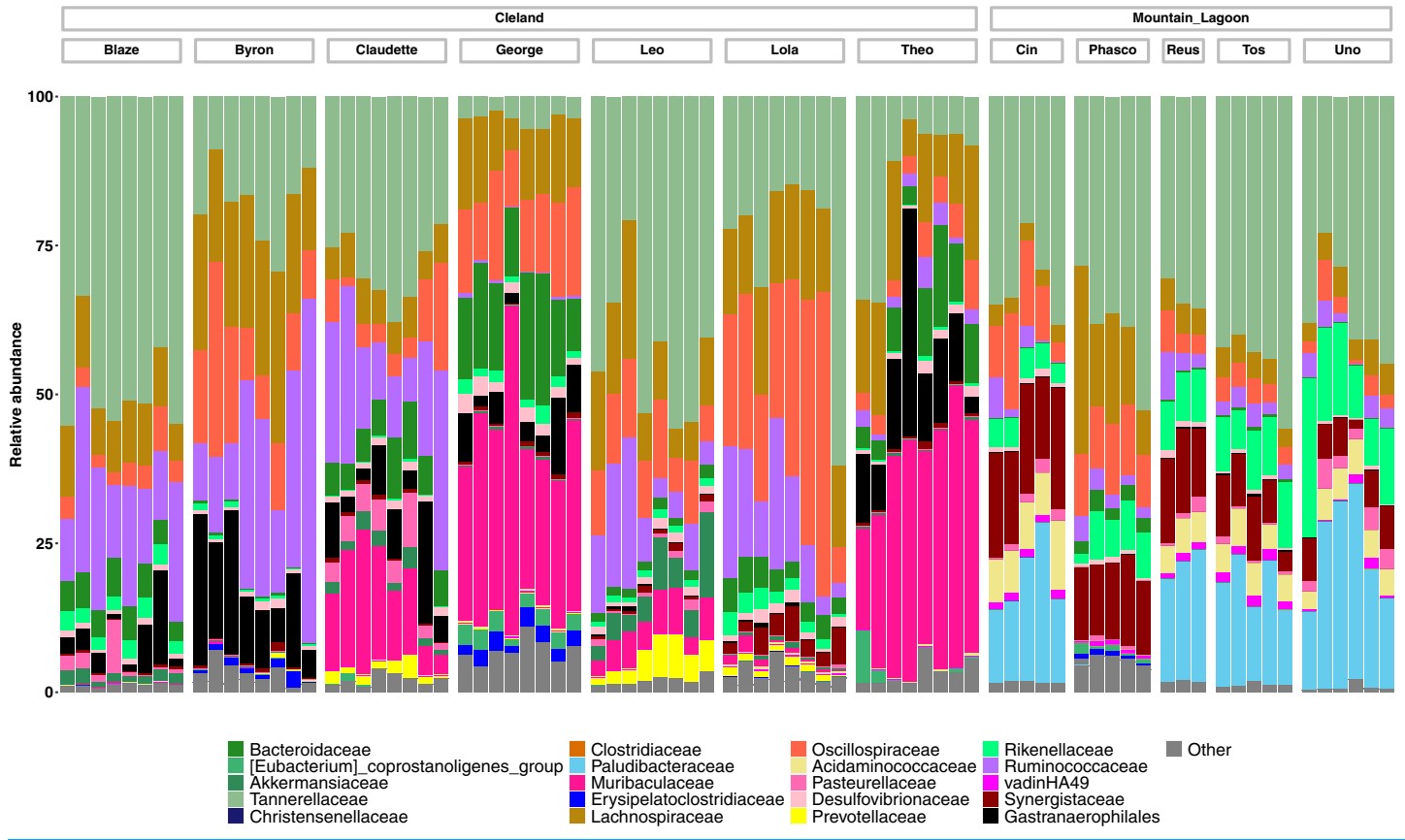

**Figure 4 Relative abundances of the 20-most abundant bacterial families in Cleland and Mountain Lagoon koalas.** Bacterial families not in the top 20-most abundant families are merged into the 'Other' bin. Individual faecal samples are grouped by koala, and ordered from first to last sample collection.

(*i.e.*, dietary and environmental stability), it is ideal to characterise this variability on the study species of interest. To this end, we sought to characterise the level of intra-individual variation in faecal microbiota composition and diversity in the koala. Both the evenness and richness of ASVs stayed relatively stable at the scale of days (Cleland koalas), and months (Mountain Lagoon koalas). We also identified a stable core microbiota within the faecal microbiota of each koala which persisted throughout the duration of sample collection. These individual core ASVs were a subset of the total diversity found in each individual (mean of 20% of the total diversity over the entire sampling of an individual) yet accounted for the majority (>95%) of the individual's microbial community in terms of relative abundance. This persistence of highly abundant individual core ASVs was observed in Cleland koalas despite their mixed *Eucalyptus* diets (though it should be noted that most of the *Eucalyptus* eaten by Cleland koalas was *Eucalyptus camaldulensis*). This suggests that while there could be an influx of transient microbes from the diet/environment, the dominant members of the koala faecal microbiota remain stable through time. These individual core ASVs were specific to individual koalas, as only a subset (five and nine ASVs) was detected in all koalas within the Cleland and Mountain Lagoon populations, respectively. This suggests that there is potentially a degree of

functional redundancy in the different individual core microbes between koalas, which has been observed in other host species (*Moya & Ferrer, 2016*; *Vieira-Silva et al., 2016*) and which could be tested in the koala in future studies using shotgun metagenomics. It is likely that these dominant individual core ASVs are inherited from the mother *via* pap—a faecal-like substance fed from the mother to the joey around the time of pouch emergence that acts as a microbial inoculant (*Osawa, Blanshard & Ocallaghan, 1993*; *Blyton et al., 2022a*). Overall, our results suggest that koalas maintain a stable, individual core faecal microbiota over the course of days to months.

We found that each individual koala had a distinct faecal microbiota composition, with inter-individual differences in composition being greater than intra-individual differences over the course of days to months. Such clustering by individuals over the course of days has been observed in desert woodrats (*Neotoma lepida*) (*Kohl, Luong & Dearing, 2015*). Long-term stability of faecal microbiota composition has also been observed in other species, albeit not to the degree that we see in koalas in this study. In chimpanzees, conserved phylotypes were detected from longitudinally sampled individuals in 2000, 2001, and 2008, and these phylotypes accounted for the bulk of the microbial community (40–70% of relative abundance) (*Degnan et al., 2012*). A degree of individuality in faecal microbiota composition over 3 weeks has previously been found in *Chlamydia*-infected koalas being treated with antibiotics in captivity (*Dahlhausen et al., 2018*). Our study extends this to healthy koalas living both in captivity and the wild, as well as increasing the sampling time to months. We found that koalas at Mountain Lagoon had more similar faecal microbial compositions when compared to koalas from Cleland. One possible explanation is that Mountain Lagoon koalas are likely more closely related and share a greater proportion of their faecal microbial communities as a consequence—especially considering the strong degree of vertical transmission in koalas (*Blyton et al., 2022a*). In contrast, Cleland koalas represent individuals from distinct and broader geographic regions (Kangaroo Island, Adelaide Hills).

Differences in faecal microbiota composition between host populations have been previously observed in other taxa, such as Black howler monkeys (*Alouatta pigra*) (*Amato et al., 2013*), and Udzungwa red colobus monkeys (*Procolobus gordonorum*) (*Barelli et al., 2015*). Such population-level differences have also been observed in the marsupial Southern hairy-nosed wombat (*Lasiorhinus latifrons*) (*Eisenhofer, Helgen & Taggart, 2021*). In this study, we observed faecal microbiota differences between the different koala populations sampled. Mountain Lagoon koalas had substantially higher relative abundance of Synergistaceae compared to Cleland koalas. Members of this family have been shown to detoxify pyridinediols and fluoroacetate in ruminants (*Allison et al., 1992*; *Davis et al., 2012*), and some have predicted that this family is important for koalas to feed on *Eucalyptus* (*Shiffman et al., 2017*). *Synergistaceae* has been found previously in koalas from Victoria and Queensland (*Brice et al., 2019*; *Shiffman et al., 2017*), so its near absence in koalas from South Australia is surprising. Another microbial family that had a much higher relative abundance in Mountain Lagoon was *Paludibacteraceae* (phylum Bacteroidota); however, little is known about this recently described family (*Ormerod et al., 2016*). *Gastranaerophilales* (phylum Melainabacteria—closely related to

Cyanobacteria) were found at a much higher relative abundance in Cleland koalas. A genome from this family has been obtained from a Queensland koala, and its putative functions involve genes relating to plant fermentation and vitamin synthesis (*Soo et al., 2014*). *Muribaculacaeae* (Formerly S24-7) was found to be highly abundant in some Cleland koalas, but completely absent from Mountain Lagoon koalas. This taxon was previously identified at high relative abundance (>20%) from captive koalas located in Queensland and NSW (*Shiffman et al., 2017*; *Blyton et al., 2022b*), and yet was also not detected from wild koalas sampled from Cape Otway, Victoria (*Brice et al., 2019*). Broader geographic sampling could determine whether this taxon is a marker for captivity, or endemic to specific koala populations. We also found *Lonepinella koalarum* to be highly prevalent (albeit low in relative abundance) across the koala samples studied. *L koalarum* has been shown to degrade tannin-protein complexes and was first isolated and described by *Osawa et al. (1995)*. This bacterium has received interest due to its positive association with koala survival during antibiotic treatment for *Chlamydia* (*Dahlhausen et al., 2018*; *Dahlhausen et al., 2020*). However, these inter-population findings should be treated with caution, as our population-level comparison is confounded by captive (Cleland) *vs* wild (Mountain Lagoon), which can influence faecal microbiota composition in some vertebrate species (*Blyton et al., 2022b*; *Kohl, Skopec & Dearing, 2014*; *Alberdi, Martin Bideguren & Aizpurua, 2021*). These comparisons are also hindered by the small individual-level sample sizes ($n = 7$ for Cleland, $n = 5$ for Mountain Lagoon). Regardless, these preliminary population-level differences are interesting—particularly in the context of adaptation to different environments and *Eucalyptus* species—and warrant follow up in future studies.

Nothing is known definitively of how differences in faecal microbial community composition are linked functionally to the consumption or digestion of individual food tree species in koalas, however associations between certain microbial taxa and koala dietary differences have been reported elsewhere. At Cape Otway, Victoria, koala diets differ in the proportions of *E. viminalis*, a highly palatable, digestible and protein-rich species, and *E. obliqua*, a less palatable and digestible species with low protein availability. These species also differ in possessing defences from different chemical classes (formylated phloroglucinol compounds *vs* unsubstituted B-ring flavanones) (*Moore & Foley, 2005*; *Marsh et al., 2021*). The faecal microbiota of koalas that feed predominately on *E. viminalis* possess more phylum Bacteroidota and less phylum Firmicutes (particularly family *Ruminococcaceae*) than those from *E. obliqua*-dominated diets (*Brice et al., 2019*). A separate study of different animals identified 25 bacterial taxa that were significantly associated with *E. obliqua* diets when compared to *E. viminalis* diet microbiota (*Blyton et al., 2019*). These were drawn from at least five phyla, however the greatest number of taxa (11) were family *Lachnospiraceae* (Firmicutes) and the sum of their mean relative abundances was >10%. The *Lachnospiraceae* are fermentative short-chain fatty acid producers, which possess strong hydrolysing potential against a variety of plant cell wall substrates (*Vacca et al., 2020*) and their association with *E. obliqua* might be explained by the high fibre content of this species presenting greater fermentative challenges. Future, larger powered studies are needed to investigate how differences in faecal microbial

community composition relate to dietary preference. Such knowledge may prove useful for conserving koalas, as determining the dietary breadth of individual koalas or populations could improve future koala translocation outcomes in Australia.

## CONCLUSION

This study found that koalas have individual core faecal microbiota that are stable over the course of days to months. This stability could be due to the lack of horizontal transfer between individuals or uptake from the environment. Such stability may have ramifications for the ability of koalas to adapt to different environments and dietary *Eucalyptus* species.

## ACKNOWLEDGEMENTS

We would like to thank the keepers at Cleland Wildlife Park for their help with collecting samples.

## ABBREVIATIONS

| | |
|---|---|
| **ASV** | Amplicon Sequence Variant |
| **CL** | Cleland Wildlife Park |
| **KI** | Kangaroo Island |
| **ML** | Mountain Lagoon |
| **NSW** | New South Wales |
| **PCoA** | Principal Coordinates Analysis |
| **PSM** | Plant Secondary Metabolite |
| **SA** | South Australia |

### Funding

Raphael Eisenhofer was funded by the Australian Research Council Centre of Excellence for Australian Biodiversity and Heritage (CABAH). Kylie L Brice was funded by the Australian Postgraduate Award (APA), Hawkesbury Institute for the Environment and Western Sydney University. Kylie L Brice was partly supported by the Glbal Centre for Land Based Innovation, also partly under the Australian Research Council's Linkage Projects funding scheme (project number LP140100751), and a Paddy Pallin Science Grant Award from the Royal Zoological Society of NSW. The funders had no role in study design, data collection and analysis, decision to publish, or preparation of the manuscript.

### Grant Disclosures

The following grant information was disclosed by the authors:
Australian Research Council Centre of Excellence for Australian Biodiversity and Heritage (CABAH).
Australian Postgraduate Award (APA).
Hawkesbury Institute for the Environment and Western Sydney University.

Glbal Centre for Land Based Innovation.
Australian Research Council's Linkage Projects Funding Scheme: LP140100751.
Royal Zoological Society of NSW.

## Competing Interests

Kellie Leigh is employed by Science for Wildlife Ltd.

## Author Contributions

- Raphael Eisenhofer conceived and designed the experiments, performed the experiments, analyzed the data, prepared figures and/or tables, authored or reviewed drafts of the article, and approved the final draft.
- Kylie L. Brice conceived and designed the experiments, performed the experiments, authored or reviewed drafts of the article, collected samples, and approved the final draft.
- Michaela D. J. Blyton conceived and designed the experiments, authored or reviewed drafts of the article, and approved the final draft.
- Scott E. Bevins conceived and designed the experiments, authored or reviewed drafts of the article, collected samples, and approved the final draft.
- Kellie Leigh conceived and designed the experiments, authored or reviewed drafts of the article, and approved the final draft.
- Brajesh K. Singh conceived and designed the experiments, authored or reviewed drafts of the article, and approved the final draft.
- Kristofer M. Helgen conceived and designed the experiments, authored or reviewed drafts of the article, and approved the final draft.
- Ian Hough conceived and designed the experiments, authored or reviewed drafts of the article, collected samples, and approved the final draft.
- Christopher B. Daniels conceived and designed the experiments, authored or reviewed drafts of the article, and approved the final draft.
- Natasha Speight conceived and designed the experiments, authored or reviewed drafts of the article, and approved the final draft.
- Ben D. Moore conceived and designed the experiments, authored or reviewed drafts of the article, and approved the final draft.

## Animal Ethics

The following information was supplied relating to ethical approvals (*i.e.*, approving body and any reference numbers):

Cleland sampling was conducted with University of Adelaide Animal Ethics Committee Approval S-2016-169 and Department for Environment and Water scientific permit Y26054.

The Mountain Lagoon project was conducted as part of a koala project with the approval of the Western Sydney University Animal Care and Ethics Committee (ACEC), reference number: A10670 and New South Wales National Parks and Wildlife Service Scientific Licence SL101364.

## Data Availability

The raw sequencing data are available at the NCBI SRA BioProject ID: PRJNA823502.

All QIIME2, code, analysis files, and R code used for analyses and to plot figures are available at GitHub: https://github.com/EisenRa/2022_Koala_Faeces_R.

## Supplemental Information

Supplemental information for this article can be found online at http://dx.doi.org/10.7717/peerj.14598#supplemental-information.

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
