# Peer review of "Individuality and stability of the koala (Phascolarctos cinereus) faecal microbiota through time"

_PeerJ, doi:10.7717/peerj.14598_

## Round 0.1 · original submission · Major Revisions

As you will see from the reviewers' comments they have raised some major issues which need to be addressed before the manuscript is resubmitted.

Reviewer 1 ·

Basic reporting

The authors assess and describe the microbiome of koalas, including longitudinal samples.

Experimental design

1) DNA Isolation: specify, if bead beating had been included or not. Normally, bead beating is advised and using or not using it changes the results.

2) Check the primer sequences published in [36]. I could find 515F and its sequence, but 806R look different (i.e., GGACTACHVGGGTWTCTAAT) and not GGACTACHVHHHTWTCTAAT.

3) Hardly ever had a publication with two different types of PCR. Do you have any cross-validation about the bias between HiFi Buffer with Platinum Taq and MyTaq HS Red Mix hot-start?

4) We set the sampling depth to 36,622 – does this mean you did random subsampling? This only adds white noise. Instead, normalize to e.g., 35,000 using simple rule-of-three.

5) Have a try with Shannon effective numbers – see publications of Lou Jost for this (https://doi.org/10.1111/j.2006.0030-1299.14714.x and https://doi.org/10.1890/06-1736.1).

6) How to you cope with spurious OTUs? Singleton removal is a bad idea, but applying an abundance cut-off is necessary. 0.25% is used in human stool sample studies, since below this value increasingly OTUs appear, which do not belong (see https://doi.org/10.1038/s43705-021-00033-z).

Validity of the findings

7) You show weighted and unweighted UniFrac (e.g., Fig. 3). Both methods have issues, either putting emphasis either on low or high abundant species, respectively (see https://doi.org/10.1186/s13059-019-1735-y for explanations). Generalized UniFrac is balanced. Especially when not taking care of spurious OTUs or not applying an abundance cut-off, this becomes an issue (see next point). In addition, “unweighted UniFrac is highly sensitive to rarefaction instance” (see https://doi.org/10.1371%2Fjournal.pone.0161196).

8) Are you sure you need to use such generic names like, e.g., ce3b37d15aee37e4213c2d13ad8b1b1a? Normally, you would have names like ASV123 or OTU77 or such. These long ASV names are very inconvenient and should be replaced. (Actually, have never seen such long generic names before).

9) You compare between both koala groups – however, you also use two different PCRs. Is it possible to do a cross-validation? You possibly still have some of the isolated DNA?

10) I am not used to pielou’s evenness for microbiome studies. According to 10.1186/s42523-020-00044-6, there is some inherent connection between species abundance and pielou’s evenness, but I still have a hard time to grasp its meaning for microbiomes. I am more used to Shannon (or Simpson) effective numbers, as said above.

11) Synergistaceae has been found previously in koalas from Victoria and Queensland, so its near absence in koalas from South Australia is surprising. --- Indeed. Please check the other studies, which primer they used! (And which DNA isolation method). Both, DNA isolation and primer, create a lot of bias. I do not hope that is the reason for your finding here! Same for the other differences.

Additional comments

12) Lines in Fig. 2 are too thick. Further, why not writing Cleland and Mountain Lagoon above the panels – the reader would immediately see which panel belongs to which study group.

13) Fig. 3 – Why not writing unweighted and weighted UniFrac above the panels and Cleland and Mountain Lagoon on the left side of the panels.

14) f = and g = are not explained.

15) Line 262: This sentence needs correction. Did you mean: “Mountain Lagoon koalas had nine ASVs that were found in all samples, of which two …“?

16) Replace =< by ≤

17) except for Phasco => except for the individual named Phasco

18) was also the only koalas to demonstrate => koala and not koalas?

Reviewer 2 ·

Basic reporting

In the present article Eisenhofer and colleagues report on the longitudinal development of gut microbiota in two small populations of captive and wild-living koalas. These are endangered marsupials and we should care about their gut microbiota, as probiotic supplementation / fecal transplants have the potential to aid in conservation efforts for this animal. The authors make a solid effort in describing a time-resolved microbiome in a total of 12 individuals. Their methods are solid and well executed, and their results are well & precisely described in writing. The main finding is that microbiota are stable over long periods of time within individual koalas, yet they differ between individuals. The major weakness that I see in the present article lies in the visualization of the data and the contextualization of previous work. I therefore suggest to make improvements in figures and discussion.

Experimental design

no comment

Validity of the findings

the authors cite literature that describes microbiota in primates, etc. I miss comparisons to articles describing the gut microbiome of koalas. Is the observed gut microbiome in these 12 animals what could have been expected?

also the PCR protocol for sequencing 16S rRNA microbiota between captive and wild koalas slightly differs. This should be noted as a bias.

Information concerning rarefaction and filtering of ASVs is missing.

method to calculate evenness and richness is missing.

Additional comments

Here are my comments on how to improve the present article:

major:

koala microbiota are stable over time, but different between individuals. Why? What could be driving this? The authors should test whether age or sex could be a confounding factor to the observed diversity. Within the captive population, there are 2 different types of koalas (Cleland & Kangoro Island) - could this be confounding?

as the microbiota are stable over time, I think they can be grouped and tested for differences between the wild and captive population. This has indeed been done, but not visualized.

that gut bacteria can influence feeding behavior in koalas is interesting. More discussion on this is necessary. Which bacteria correlate with certain eucalyptus species? Is there something known about this? I feel like this should be much more central to the discussion - instead lines 303 to 328 and lines 350 - 370 are more like repetitions of results. I would like to see the results contextualized with relevant and recent literature concerning the following questions:
How will understanding koala - microbiome help us conserve koalas with respect to diet?
How will understanding koala - microbiome help us conserve koalas with respect to chlamydia infection?
How does this marsupial microbiome relate to other marsupials + to other mammals (focus on the concept of phylosymbiosis).

I feel like this article would be heavily improved if it would start with Figure 4 (+genus level as supplementary). This gives a good insight to start with & intuitively raises the question to why we observe differences in individuals -> which can then be answered in the following Figures. Paludibacter seems prevalent in the wild population - why is that? Much of this should be added to the discussion.

Figure 2 is not informative, but visualizing the comparison of within-group evenness/richness between wild and captive would be informative.

Figure 3 would be highly improved if the coloring would not be based on individual but on wild or captive.

---

## Round 0.2 · Minor Revisions

Reviewer 2 has noted a few minor issues that should be addressed before the manuscript is accepted. These are problems with Figure 4, a need to address PCR bias, and missing information about filtering of ASVs

Reviewer 1 ·

Basic reporting

The authors assess and describe the microbiome of koalas, including longitudinal samples. All points of my last review have been sufficiently addressed, even though there still is same disagreement about some points. However, this manuscript is about koalas and not about issues on how to handle 16S data.

Experimental design

see above

Validity of the findings

see above

Additional comments

As said, there is no need for further changes in the manuscript (i.e, manuscript can be published as is), since everything is justified. However, for (future) clarification, I would like to address the following:
1) Rarefaction means random subsampling of reads. This random process only adds noise, which is not necessary. Instead, all samples could be normalized to a fixed value by using simple math (rule of three, rule of proportion - at least that is what my dictionary tells me it is told in English ...). If you do rarefaction many times and you take the mean of all rarefaction runs, you will see that the values converge (with increasing numbers of rarefaction runs) to the exact same values you would have gotten by simple math in the beginning. I know that many pipelines still use rarefaction, but 'usage of many' does not make it better. Sometimes I even have the suspicion rarefaction became only popular, because it is more 'fancy' compared to simple normalization using division and multiplication. I argue that rarefaction is inferior. Since you are using a high read value of >30,000 reads, the issue of rarefaction is less pronounced. It becomes a problem if people have a sample with a low number of reads and rarefy towards this low number.
2) Concerning UniFrac, I was suggesting to use generalized UniFrac, which is balanced in comparison to either weighted or unweighted UniFrac.
3) Pielou: The definition was clear, but what does this value means for the interpretation of a microbiome? I am still not sure about what does Pielou is telling me about the individual koalas or in comparison between the animals.
4) 0.25% abundance value cut-off => keeping only those OTUs occurring at a minimum relative abundance of 0.25% in at least one sample (in the said publication, page 7, left column, near the end of page).

Reviewer 2 ·

Basic reporting

I had two major concerns for the original manuscript:
1) The structure and visualization of provided information
2) The difference in PCR protocols for sequencing 16S rRNA microbiota between captive and wild koalas.

Both of my concerns were partially addressed.

- More context is now provided (citing other literature, expanding on discussion...).

- The structure of Figures remains the same. While I recommended to focus on a comparison between captive and wild-living koalas, the authors insisted on depicting microbiota from individual samples. This description is still valuable. I still think that the finding is interesting - that summarized alpha diversity is obviously lower in one compared to the other group, and think it should be highlighted as a main Figure. But I comply.

- Figure 4 has no x-axis and typos in the figure legend. Also, the fact that 20 families account for 98,1% rel.abundance is a result and should not be in the legend.

- The authors note to have addressed missing information about filtering of ASVs (e.g. rare ones) in the revised method section. I could not find this revised information.

- The authors note the differences of employed polymerases in their study. However, potential bias is not only introduced because of the difference in polymerase. There are substantial differences in PCR cycle number, annealing temperature, etc. These differences remain not addressed.

Experimental design

bias in PCR

Validity of the findings

filtering of very rare and spurious taxa is often necessary to establish stable results. It is unclear to me whether this has been done and how it translated into calculations of alpha and beta diversity.

Additional comments

-

---

## Round 0.3 · accepted · Accept

I am happy that you have addressed all the reviewers' comments, I have assessed the revision and am happy that it is now accepted for publication.